# Ice volume and climate changes from a 6000 year sea-level record in French Polynesia

N. Hallmann [1], G. Camoin[1], A. Eisenhauer[2], A. Botella[3], G.A. Milne[3], C. Vella[1], E. Samankassou [4], V. Pothin[1], P. Dussouillez[1], J. Fleury[1] & J. Fietzke [2]

Mid- to late-Holocene sea-level records from low-latitude regions serve as an important baseline of natural variability in sea level and global ice volume prior to the Anthropocene. Here, we reconstruct a high-resolution sea-level curve encompassing the last 6000 years based on a comprehensive study of coral microatolls, which are sensitive low-tide recorders. Our curve is based on microatolls from several islands in a single region and comprises a total of 82 sea-level index points. Assuming thermosteric contributions are negligible on millennial time scales, our results constrain global ice melting to be 1.5–2.5 m (sea-level equivalent) since ~5500 years before present. The reconstructed curve includes isolated rapid events of several decimetres within a few centuries, one of which is most likely related to loss from the Antarctic ice sheet mass around 5000 years before present. In contrast, the occurrence of large and flat microatolls indicates periods of significant sea-level stability lasting up to ~300 years.

[1] Aix-Marseille Université, CNRS, IRD, Collège de France, CEREGE Europôle Méditerranéen de l'Arbois, BP80, 13545 Aix-en-Provence Cedex 4, France. [2] GEOMAR Helmholtz-Zentrum für Ozeanforschung, Wischhofstraße 1-3, 24148 Kiel, Germany. [3] Department of Earth and Environmental Sciences, University of Ottawa, Ottawa, ON K1N 6N5, Canada. [4] Department of Earth Sciences, University of Geneva, Rue des Maraîchers 13, CH-1205 Geneva, Switzerland. Correspondence and requests for materials should be addressed to N.H. (email: hallmann@cerege.fr)

Current and future changes in the Earth System pose some of the most pressing scientific and societal issues—how will climate, the ocean and ice sheets respond to warming from greenhouse gas forcing? Global mean sea level is projected to rise a few decimetres to over a metre by the year 2100[1]. Sea-level rise will have severe impacts on coastal ecosystems, water supplies and densely populated coastal areas, including low-lying islands, which are usually considered as one of the most vulnerable world regions under future sea-level rise[2]. However, centennial to millenial projections of sea-level rise remain highly uncertain, primarily due to our relatively poor understanding of the sensitivity of the ice sheets to sustained warming and of the spatial and temporal sea-level variability on these time scales.

Reconstructions of past sea-level changes help in predicting the response of the climate system to human-induced perturbations by defining the backdrop of natural variability. Most instrumental observations span only the past few decades and so confidently separating the anthropogenic and natural signals requires the use of geological archives that capture changes on climate-relevant time scales. Mid- to late-Holocene records provide an important baseline prior to the industrial revolution[3] through the use of various sea-level indicators including salt marsh microfossils, archaeological indicators, vermetid constructions and fossil corals[4,5].

Microatolls are coral colonies with living outer margins and flat dead upper surfaces that have grown laterally for decades or centuries as their vertical growth has been constrained by exposure at lowest tides[6–8]. They can be considered as reliable indicators of the mean low water springs (MLWS) level in open water settings, although some of that variability may be related to subtle hydrodynamic conditions[8]. Coral microatolls therefore represent unique archives to reconstruct regional Holocene sea-level trends[5,9] and climatic oscillations[10] on interannual to millenial time scales.

In low-latitude regions, away from former and present ice sheets (so-called 'far-field' regions), reconstructed sea-level changes provide the most accurate and precise estimate of ice volume changes once they have been corrected for the influence of tectonic processes and glacial isostatic adjustment (GIA)—the deformational, gravitational and rotational response of the Earth to the ice-ocean mass exchanges associated with the glacial cycles[11,12]. In contrast to typical mid-to-low latitude Atlantic sea-level curves that indicate continuously rising sea levels at a decelerating rate during the Holocene[13], far-field RSL records often display a highstand during the mid- to late-Holocene in the Pacific[14–17] and in the eastern Indian Ocean[18,19]. This reflects a reduction in the rate of global melt water production and the subsequent dominance of GIA processes. Specifically, the local influence of water loading (known as hydro-isostasy[11]) and the global influence of ice- and ocean-induced loading on ocean basin volume (termed ocean syphoning[20]) led to a fall in sea level. Given the relatively small ice volume contribution to RSL change during the mid- to late-Holocene[21], inference of this signal requires an accurate estimate of the GIA contribution as well as precise reconstructions of RSL, both of which are key aspects of this study.

Here, we reconstruct mid- to late-Holocene RSL changes on a century time scale based on high-precision GPS positioning and U/Th dating of coral microatolls from five islands in a single region. French Polynesia is an ideal region for palaeo sea-level studies due to its location in the far field, where accurate information regarding the melting history of ice sheets can be extracted from sea-level archives, and to its tectonic stability over the studied time window. We document a sea-level highstand of less than a metre between 3.9 and 3.6 kyr BP, thus giving a glacio-eustatic contribution of 1.5–2.5 m since 5.5 kyr BP, most likely

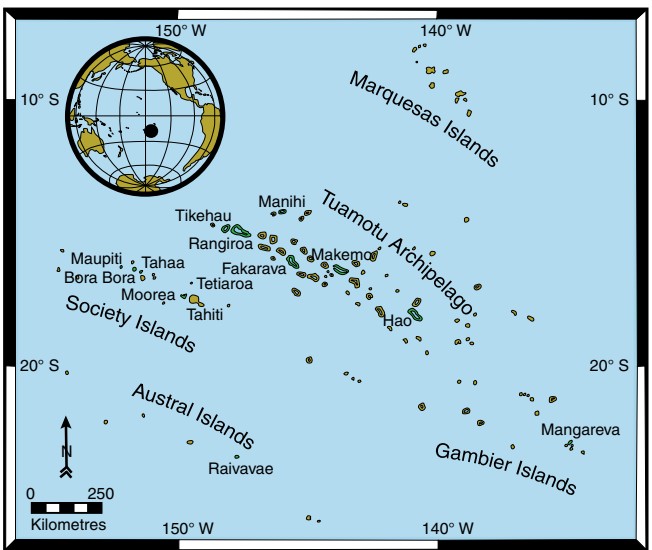

**Fig. 1** Map of French Polynesia. Modern and Holocene reef systems have been studied on 12 islands from four archipelagos (Tuamotu, Society, Gambier and Austral islands) in French Polynesia

sourced from the Antarctic ice sheet. We also demonstrate the occurrence of isolated and relatively rapid decimetre-scale RSL changes as well as periods of significant sea-level stability lasting up to ~300 years over the last 6000 years.

## Results

**Mid- to late-Holocene sea-level record in French Polynesia.** Mid- to late-Holocene reef sequences have been studied on 12 islands from four archipelagos in French Polynesia (Fig. 1 and Supplementary Table 1), which cover a wide range of latitudes and longitudes (S14°20′ to S23°54′, W152°18′ to W134°50′).

These islands represent ideal settings for accurate palaeo sea-level studies based on geographic, tectonic and oceanographic parameters. They are far-field sites and therefore have the potential to provide accurate information regarding the melting history of ice sheets. The age of the Society Islands decreases from the NW to the SE, as a consequence of the Pacific plate motion over a fixed hot spot[22]. These islands experienced, accordingly, a slow and regular subsidence during the late Pleistocene and the Holocene, especially due to the cooling and the concomittant sinking of aging lithosphere.

As consistently assessed by several approaches, subsidence rates range from 0.05 mm per year in the northwestern part of the Society Islands[23] to $0.15 \pm 0.15$ mm per year in Tahiti, where five independent geophysical measurements were used[24]. Tectonic corrections that have been applied to the relevant studied islands do not exceed 25 cm and are not more than 15 cm for the time window encompassing the mid- to late-Holocene sea-level highstand. The studied Tuamotu Islands, Tikehau and Rangiroa, formed between 47 and 55 Ma as evidenced by palaeontologic and radiometric age data[25,26]. These two atolls are considered as tectonically stable since at least the late Quaternary based on the elevation of the Last Interglacial reef terrace (+6 m[27]), which is in agreement with estimates of global mean sea level during this period[28]. Even though departures from global mean sea level at this location during the Last Interglacial could have been several metres[29]; the inferred mean tectonic rate remains relatively small (order $10^{-2}$ mm per year). This demonstrates that the uplift due to the deflection of the crust under the Tahiti volcanic load[27] ceased before the formation of this terrace.

All studied islands are characterised by a low tidal amplitude ranging from less than 0.15 m in the Society Islands up to 0.5 m in the Gambier Islands, with most values lying between 0.2 and 0.3 m (Supplementary Fig. 1). Such a low tidal amplitude allows a greater accuracy and precision in reconstructing sea level by, for example, lowering the potential effects of ponding at low tide that can introduce a significant bias.

The mid- to late-Holocene reef sequence is up to 2 m thick and consists of distinctive terraces comprised of in situ coral microatolls, which locally coalesce to form compact fields. The microatoll terraces are unevenly covered by thick accumulations of conglomerates composed of reworked coral colonies, and corresponding to storm deposits (Fig. 2).

**Mid- to late-Holocene sea-level curve.** Our sea-level reconstruction is based on U/Th dating of in situ coral microatolls and their precise GPS positioning (see 'Methods' section; Supplementary Tables 1 and 2).

At our studied sites (Fig. 1), modern adjacent microatolls growing in the same environment and under similar hydrological conditions reveal a difference in their elevation of 5 cm on average, which can be considered therefore as the resolution at which coral microatolls can resolve sea-level changes in a specific reef environment. The height of living microatolls (biological level, see 'Methods' section) varies in different reef environments (Fig. 3): reef flat, lagoonal areas and shallow waterways (hoas) through islets (motus), which do not connect permanently the

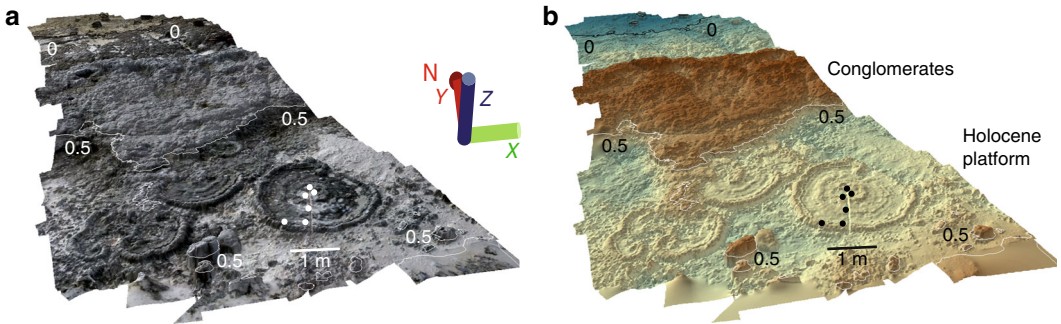

**Fig. 2** Photogrammetry for a study site in Maupiti. 3D reconstruction with **a** photographic texture and **b** coloured texture indicating heights. The numbers indicated in the figure correspond to elevations in metres. Microatolls are located on the Holocene platform and partly covered by conglomerates. One of these microatolls was slabbed for palaeoclimate analysis. Circles indicate position of the samples MAU-97–102. The scale changes due to the perspective view. Agisoft Photoscan Professional version 1.2 was used to produce the 3D model

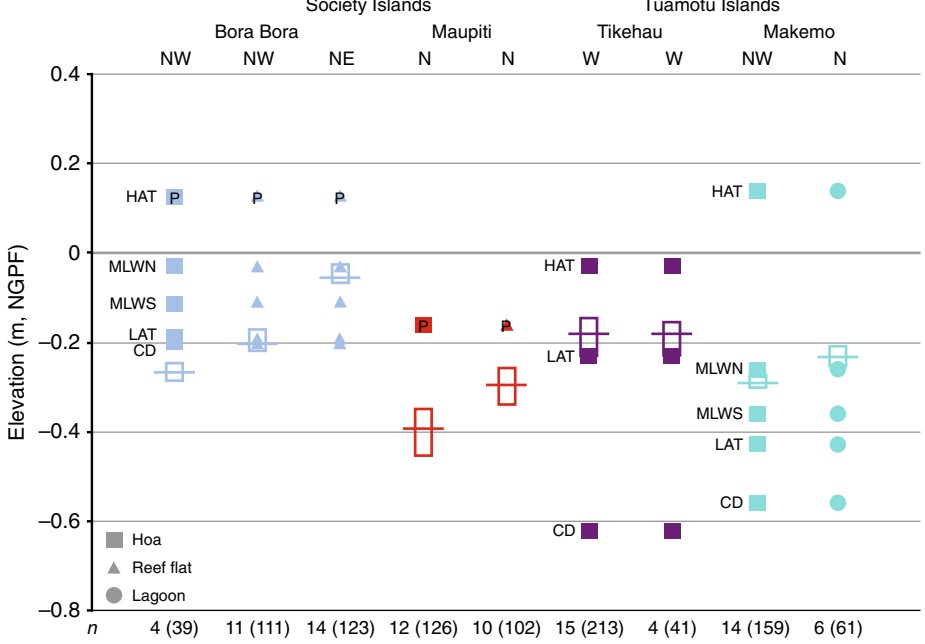

**Fig. 3** Biological levels of modern microatolls in different depositional environments compared with tidal datums. The tidal datums are related to the NGPF. The open boxes represent the surveyed microatoll elevations at different sites from four islands, and the horizontal line indicates the mean biological level (MBL, see text). HAT, highest astronomical tide; MLWN, mean low water neaps; MLWS, mean low water springs; LAT, lowest astronomical tide; CD, chart datum. P, probe measurements (this study). See 'Methods' section for details on GPS microatoll positioning and probe measurements. The respective numbers ($n$) of microatolls and related GPS measurements (in brackets) per studied site are shown at the bottom of the figure

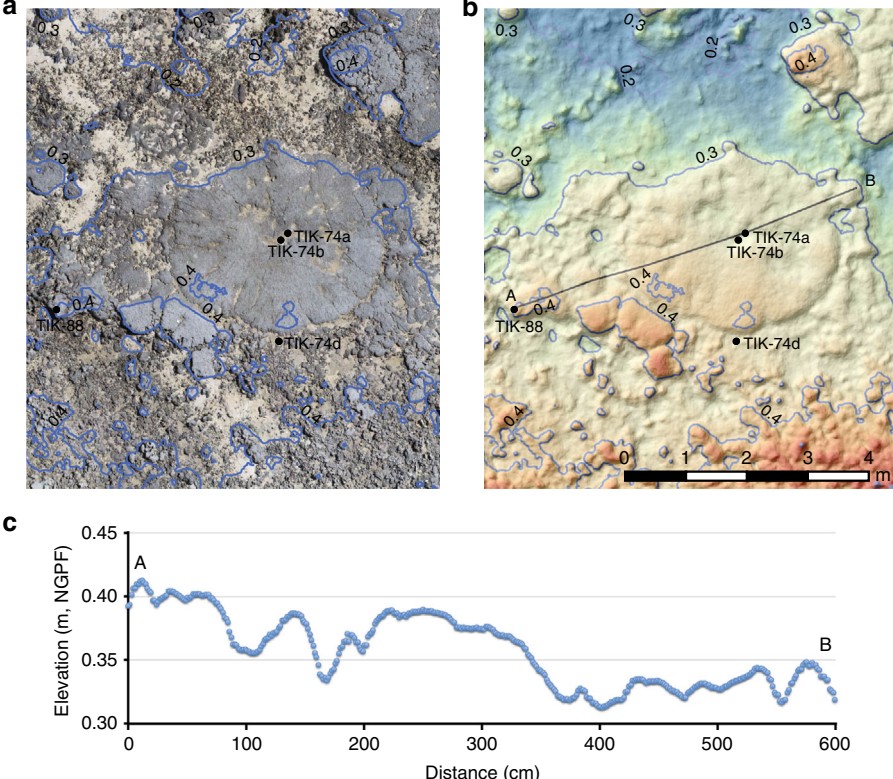

**Fig. 4** Photogrammetry for a large microatoll from Tikehau. Variations of the surface elevation of a microatoll with a diameter of 8 m indicate low-amplitude, high-frequency sea-level variations on the order of 10 cm from 5.59 ± 0.05 kyr to 5.41 ± 0.08 kyr BP. 3D reconstruction with **a** photographic texture and **b** coloured texture indicating heights. Variations of the surface elevation of TIK-74 are shown in **c**. The numbers indicated in the figure correspond to elevations in metres. Agisoft Photoscan Professional version 1.2 was used to produce the 3D model

open ocean to the lagoon[30]. In the studied islands, it averages 18 cm below mean sea level (bmsl) in reef flat environments, 23 cm bmsl in lagoonal areas and 26 cm bmsl in hoas. This reflects a variability in sea level at various time scales that may be related to local hydrodynamic conditions in specific environments or to short-term changes in water level of a few decimetres induced by winds and/or swells[31,32]. A significant variability may be induced by moating effects in subenvironments that are not freely connected to the open ocean and where microatolls can survive at elevations substantially above contemporaries lower in the tidal range[6,33]. These processes prevail in areas characterised by a macrotidal regime[33] but are limited in areas where the tidal range is minimal, such as the studied islands, implying that the sea-level database used in this study is reliable.

Seventy-two *Porites* microatolls occurring at elevations of −0.03 to +0.65 m relative to the present mean sea level (MSL) have provided U-Th ages ranging from 5.59 ± 0.05 to 1.26 ± 0.02 kyr BP (Supplementary Tables 1 and 2), thus encompassing most of the mid- to late-Holocene time window. In addition, the dating of in situ flat coral colonies helps to further constrain the sea-level curve as the morphology of these colonies results from the influence of the tidal regime and may correspond to an early microatoll development stage[7].

The storm deposits which overlie microatoll terraces yielded ages ranging from 5.1 to 1.5 kyr BP with a maximum occurrence between 2.5 and 1.5 kyr BP, indicating a sustained storm activity during this time window, in good agreement with data from Tahaa island[34].

Individual microatolls range in diameter from 0.4 to 8 m and are generally characterised by their overall flat surface ('microatoll plane') as a consequence of periodic subaerial exposure at low

tide[7]. This indicates that sea-level variations were not large enough (i.e., within 20 cm) to displace the optimal zone for microatoll growth or to alter significantly their growth morphology. These microatolls therefore typify sea-level stillstands on decadal to centennial time scales. Most microatolls average 1–1.6 m in diameter and therefore represent a 50–80 years time span based on the average growth rate (10 mm per year) of modern *Porites* colonies in French Polynesia[35]. Significantly larger microatolls (Fig. 4), up to 6 m in diameter, crop out in Bora Bora, Maupiti and Tikehau and characterise longer sea-level stillstands (up to three centuries) during four specific time windows (Fig. 5).

Mid- to late-Holocene RSL changes have been reconstructed using two independent methods (see 'Methods' section): (1) the direct comparison between elevations of fossil microatolls and their living counterparts growing in the same depositional environment (Fig. 5a), and (2) the GPS measurements of microatoll elevations referenced to local tidal parameters (Fig. 5b) defined by the RGPF (geodetic datum in French Polynesia). In both cases, the microatoll elevations have been corrected for vertical tectonic movements (GIA corrections are applied in the following section). In the first approach, which is the more accurate and precise of the two, it was assumed that the relationship between microatolls and the tidal cycle has remained the same during the relevant time interval; thus, the reconstructed sea-level curve reflects relative changes of the MLWS level throughout the mid- to late-Holocene compared to its modern position (Fig. 5a).

The oldest dated microatolls from two sites in Tikehau belong to fields occurring at a present elevation of 40–50 cm above mean

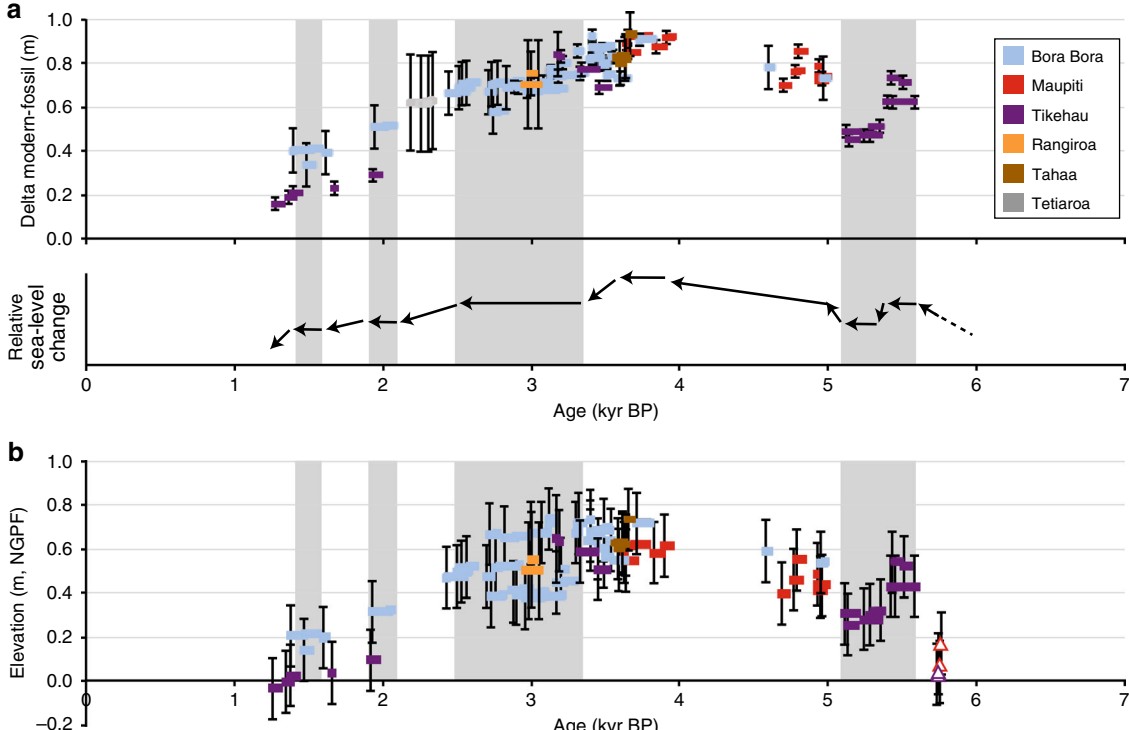

**Fig. 5** Mid- to late-Holocene sea-level curve for French Polynesia. Age vs. elevation plot of dated microatolls (solid symbols) and reef flat facies (open symbols); GIA corrections were not applied to the data. Calculated elevations are based on **a** the height difference between living microatolls and their fossil counterparts ('Delta modern-fossil') and **b** NGPF measurements (geodetic datum in French Polynesia, see 'Methods' section). Error values for ages and elevations are 2-sigma (see 'Methods' section and Supplementary Table 1). Relative sea-level changes are indicated by the direction of the arrows. Horizontal bars illustrate large microatolls, which developed during periods of relative sea-level stillstands. The grey shaded areas indicate specific time windows characterised by these century-scale stillstands. Data concerning Tetiaroa were obtained during a parallel study (A.E., unpublished)

sea level and yielded ages ranging from 5.59 ± 0.05 to 5.43 ± 0.04 kyr BP, thus indicating a MLWS level of +60 to 70 cm during this time interval. A sea-level rise averaging 1.5–1.75 mm per year is therefore documented between 6 kyr BP, when its position was close to the present level[36], and 5.5 kyr BP. A slight drop in sea level is evidenced by the development of a microatoll field at a present elevation of 30 cm above mean sea level in Tikehau and dated from 5.37 ± 0.05 to 5.12 ± 0.04 kyr BP, characterising a MLWS level of +50 cm during this time window. Moreover, the occurrence of very large microatolls in this field (Fig. 4) indicates a stable sea-level position for several centuries after the abrupt drop. Microatolls from western Australia also indicate an abrupt fall at the same time[37]. A number of Atlantic paleoclimate records indicate the occurrence of a cooling transition between 6 and 5 kyr BP[38] that has been linked to changes in total solar irradiance[39]. We speculate therefore that the RSL fall was driven by cooling and ice mass increase in the North Atlantic region[39]. There may also have been a cooling of Pacific waters and thus a steric contribution to the RSL fall as sea-surface temperature (SST) variability of about 1 °C has been reported in some parts of the Pacific[40].

To quantify the possible influence of steric changes on RSL in the studied region, we examined output from three climate models (see 'Methods' section). This output indicates that sea-surface height (SSH) changes are correlated to SST changes over a large range of time scales. For example, correlation coefficients of 0.8–0.97 for a 100-year running mean applied to time series of SSH and SST demonstrate a strong correlation over century time scales. At Bora Bora, a variation of 1 °C in SST is associated with a variation of 7–10 cm in sea level. For comparison, El Niño events result in SST and SSH variations of several degrees C and several

decimetres[41] and have shown fluctuations over the Holocene[42]. A significant steric contribution to the isolated, rapid RSL fall is therefore plausible.

A cluster of points from Maupiti and Bora Bora represent microatoll fields which crop out at present elevations of 30–35 cm, indicating that the MLWS would have been located at +70 to 80 cm between 5 and 4.6 kyr BP. A sea-level rise of a few decimetres at a rate of several mm per year is therefore evidenced between 5.1 and 5 kyr BP. As these corals were collected on different islands, a GIA correction was applied and accurately delineated a sea-level rise of 20–43 cm at rates of 1.2–2.5 mm per year (see 'Methods' section). A rise of similar amplitude is also documented in microatolls from the western Pacific[37] A spike in Antarctic ice-rafted detritus around 5 kyr BP[43] indicates a possible southern mass contribution to this rise, perhaps related through changes in Atlantic circulation[39].

Our dataset demonstrates that the MLWS level was at +0.9 m between 3.9 and 3.6 kyr BP (Fig. 5a), inducing the development of widespread microatoll fields displaying large colonies of more than 1.6 m in diameter, especially in Maupiti and Bora Bora. However, the start of the sea-level highstand could not be precisely defined due to a lack of data between 4.6 and 3.9 kyr BP. The reported rates of sea-level rise in the 5.5–3.9 kyr BP time window range from ~0.1 to ~3 mm per year and are therefore in the lower range of the reported sea-level variability from the previous interglacial period (1–7 mm per year on average, based on last interglacial sea-level changes expressed in m/kyr)[28]. The inferred timing and magnitude of the Holocene sea-level highstand in French Polynesia significantly differ from previous reconstructions concerning this region[14,17], which were based on few accurate sea-level indicators, i.e., three microatolls from two

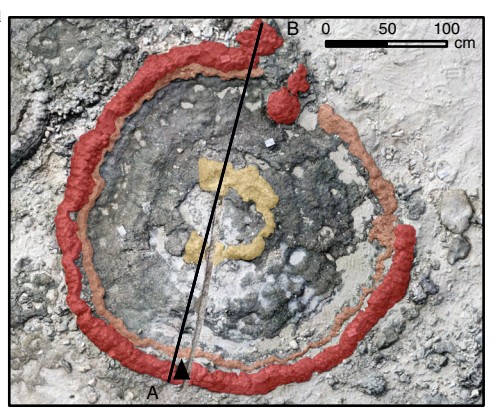

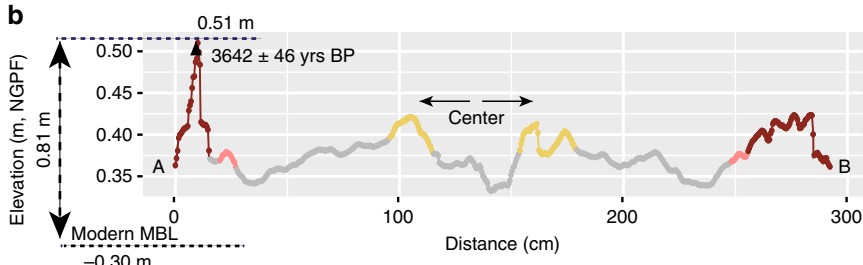

**Fig. 6** High-frequency sea-level fluctuations. Variations of the surface elevation of a microatoll from Maupiti indicate low-amplitude, high-frequency sea-level fluctuations on the order of 20 cm. **a** Photogrammetry and **b** variations of the surface elevation of MAU-101. The central part of the microatoll and its outer margins are highlighted in yellow and red, respectively. The triangle indicates the position of the dated sample. The modern counterpart is located at −0.30 cm and reveals a 0.81 m higher sea level at 3.64 ± 0.05 kyr BP. The arrows indicate the coral growth directions. Agisoft Photoscan Professional version 1.2 was used to produce the 3D model

islands[17] and six microatolls from six islands[14]. Our results exclude a sea-level highstand of more than 1 m as indicated in an earlier study (estimated at a minimum of +1.5 m[17]), which could be related to the lack of use of precise GPS measurements and the dating of reworked coral colonies. Furthermore, our results also contradict a long-lived Holocene sea-level highstand that was previously considered (i.e., 5–1.5 kyr[14] and 5.4–2 kyr BP[17]). Although field evidence on continental margins does support highstands of greater amplitude[44] as a consequence of hydro-isostasy[11], the occurrence of such large amplitude sea-level change on mid-ocean islands from the far field is more difficult to explain.

A sea-level fall, at an average rate of ~0.3 mm per year, is recorded between 3.6 and 1.2 kyr BP when sea level was about a decimetre from its present position (Fig. 5a) and which consists of the youngest microatoll record in the study area. Our results are in clear contradiction with the occurrence of a short-lived sea-level peak of 1 m in amplitude between 2 and 1.5 kyr BP[14], which was probably deduced from the dating of coral colonies reworked in storm deposits that commonly overlie the microatoll fields. The occurrence of large microatolls of more than 3 m in diameter in two specific time windows, 2.9–2.7 and 1.6–1.4 kyr BP, indicates that sea level did not fall at a continuous rate, but was characterised by significant variability. These relative sea-level stillstands indicate extended periods during which the net contribution to local sea-level change was close to zero. These could be periods during which the glacio-eustatic sea-level rise was large enough to counteract the regional GIA contribution (dominated by syphoning), suggesting significant century-scale variability in ice volume changes.

The large microatolls often exhibit a complex microtopography of their upper surface typified by the occurrence of ridges and different generations of overgrowths, generally 5–10 cm high ('Multiple-ringed' microatolls[7]), reflecting high-frequency, low-

amplitude sea-level variations on decadal to sub-decadal time scales that occurred over the coral lifetime (Fig. 6). Such sea-level variability could have been driven by the El Niño-Southern Oscillation, which has a strong impact in the western tropical Pacific with sea-level highs and lows of 20–30 cm compared to normal conditions[1,45]. Sea-level variations of similar amplitude and frequency have been reported in mid- to late-Holocene salt marsh deposits[4]. This demonstrates that microatolls are sensitive recorders of different modes of sea-level variability at various time scales, including subtle sea-level variations.

**Geophysical modelling.** Our French Polynesian record displays striking differences with sea-level data obtained on nearby archipelagos. On the Cook Islands, which are characterised by low subsidence rates (~0.01 mm per year[9]), a sea-level highstand of at least 1 m[46] and up to 1.5 m[47]–1.7 m[48] has been reported between 5.14 and 3.62 kyr BP. Studies of microatolls from Christmas Island[5] reported fairly stable RSL with oscillations not greater than 0.50 m in amplitude over the last 6000 years (Supplementary Fig. 2). To explore these regional differences and extract ice volume changes from our new sea-level curve, we apply a geophysical model of the GIA process.

To estimate the contribution of GIA to local changes in RSL, a model calibration was performed to identify a set of parameters that produce optimal fits to the dataset within the parameter ranges explored (see 'Methods' section). To this end, two ice models, ICE-5G[49] and a recently published model, BM2016[50], were considered in combination with a spherically symmetric Earth model with varying viscous properties. Model outputs using viscosity parameters that produce optimal fits to the recon-structed curve (see 'Methods' section) are compared to the data in Fig. 7; these outputs illustrate the contribution of GIA to the temporal variability of sea-level changes at the millennial scale.

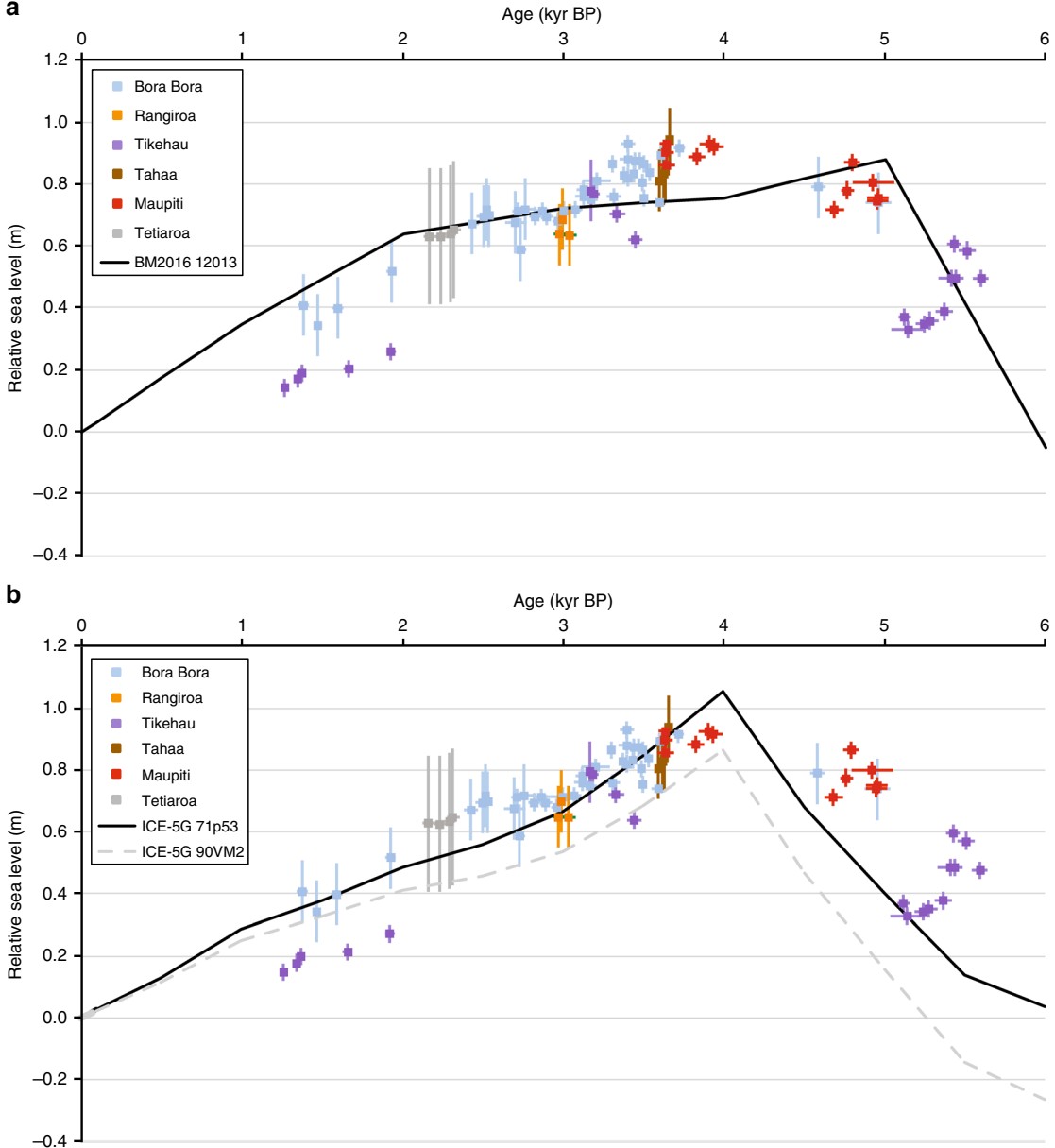

**Fig. 7** Comparison of observations and predictions of sea-level change for French Polynesia. **a** Reconstructed RSL compared to model output (black line) based on the BM2016 ice model and the viscosity model that optimised the data-model fit (see main text). In order to remove spatial variability associated with GIA, data were corrected to the location of Bora Bora using these model parameters (see 'Methods' section). **b** Same as in **a** except that the ICE-5G ice model was used with two viscosity models; one that optimised fits to the French Polynesian data (see main text; black line) and the VM2 model (dashed grey line[50]). Error values for ages and elevations are 2-sigma (see 'Methods' section and Supplementary Table 1)

The data-model fits produced by each ice model and its optimal viscosity model are equivalent at the 95% confidence level (based on an $F$-test). Thus, given uncertainty in Earth viscosity structure, the data are unable to distinguish between the two ice histories considered. The viscosity profiles that minimised the data-model misfit are: (1) for BM2016; lithosphere thickness (LT) of 120 km, upper mantle viscosity (UMV) of $10^{21}$ Pa.s and a lower mantle viscosity (LMV) of $3 \times 10^{21}$ Pa.s; (2) for ICE-5G; LT = 71 km, UMV = $5 \times 10^{20}$ Pa.s and LMV = $3 \times 10^{21}$ Pa.s. Although each ice model is able to fit the data equally well, the optimal model curves (black lines in Fig. 7) are different in form and neither model captures the data well over the entire ~6 kyr period. The BM2016 model captures the older data better than ICE-5G as it produces an earlier highstand; however, the curve produced by

ICE-5G better tracks the observations for the period after 4 kyr BP. These differences reflect the glacio-eustatic functions that are implicit in each ice model (Fig. 8). Following the cessation of significant melting in each ice model, the rate of RSL fall is close to 0.3 mm per year. This rate is consistent with observations from nearby archipelagos like Cook Islands[9] and is typical for mid-ocean, far-field regions of the South Tropical Pacific, where RSL changes are dominated almost exclusively by ocean syphoning.

As previously discussed, the observations show variations at century time scales as well as some degree of vertical scatter that may primarily reflect disparities in environmental conditions, but also the existence of decimetre amplitude sea-level fluctuations associated with processes (e.g., climate modes; rapid ice volume fluctuations) that are not reproduced in the GIA model.

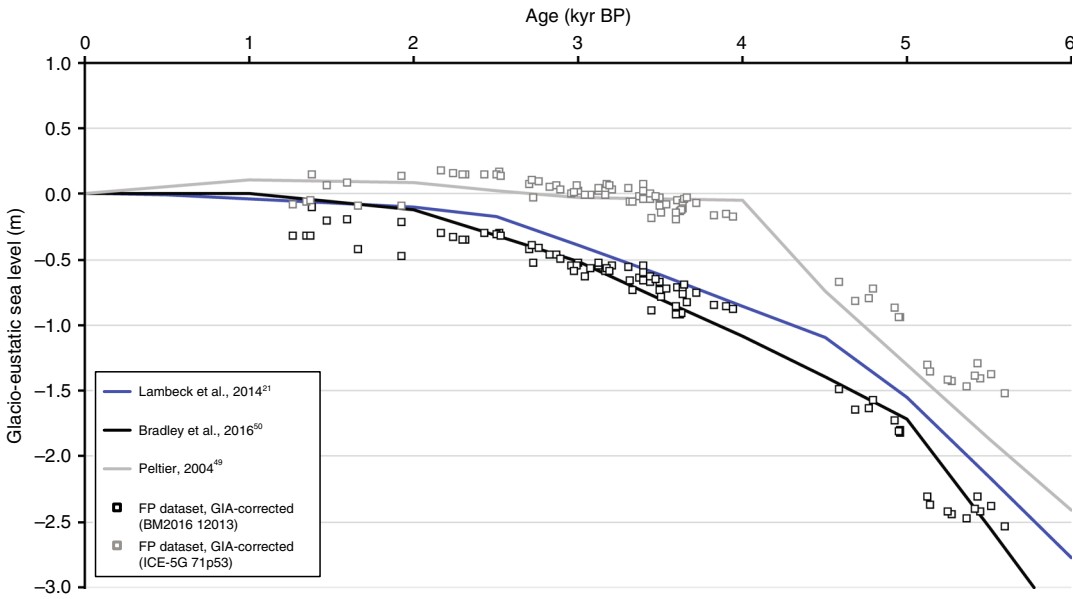

**Fig. 8** Comparison of glacio-eustatic sea-level change estimates. Glacio-eustatic sea-level changes (changes in ice volume only) as represented in BM2016 (in black) and ICE-5G (in grey). Observations (black and grey squares) corrected for GIA based on the best-fitting model (in black: BM2016; LT = 120 km; UMV = $10^{21}$ Pa.s; LMV = $3 \times 10^{21}$ Pa.s; in grey: ICE-5G; LT = 71 km; UMV = $0.5 \times 10^{21}$ Pa.s; LMV = $3 \times 10^{21}$ Pa.s). Glacio-eustatic history inferred by Lambeck et al.[21] is shown in blue

In order to estimate changes in global ice volume during the mid- to late-Holocene, the above-mentioned optimal models were used to calculate GIA signals that were then removed from the dataset. The residual reflects sea-level changes due to ice volume changes or regional changes in ocean properties leading to a steric sea-level component (Fig. 8). The residuals plot close to their associated model glacio-eustatic curve, reflecting the influence of ocean loading on the computed GIA response. This also illustrates an element of circularity in inferring an ice volume (or steric) signal using a model correction. As neither ice model was preferred by the observations when uncertainty in Earth viscosity structure was considered, it is not possible to select between the two glacio-eustatic curves inferred from the model-corrected observations (boxes in Fig. 8). The determination of a more precise constraint on ice volume change using the sea-level data presented here will require independent constraints on regional viscosity structure, as well as an estimate of a potential steric overprint. Regarding the latter, centennial to millenial changes in SST on the order of 1 °C have been observed in various places of the Pacific[51] during the study period, implying changes in steric sea level on the order of 10 cm. However, no temperature data are available for the sites considered in this study.

As noted in the RSL data-model fits, the BM2016 model compares well to the older GIA-corrected data while the ICE-5G results track the GIA-corrected data better after 4 kyr BP; this is also reflected in the consistency between the model curves (black and grey lines) and the model-corrected data (black and grey boxes) in Fig. 8. We include, for comparison, the glacio-eustatic model inferred from a recent global analysis[21] of far-field RSL observations. Based on the results shown in Fig. 8, and assuming no secular steric overprint, we infer a reduction in global grounded ice volume of ~1.5 m to ~2.5 m since ~5.5 kyr BP. As noted above, the uncertainty in this range reflects the differences between glacio-eustatic curves in the adopted ice models rather than uncertainty in the RSL reconstructions. Given that volume changes in the Greenland ice sheet are thought not to exceed a few decimetres over this period[52], the Antarctic ice sheet was likely the dominant source of this melt water. Field constraints on the retreat chronology of the Laurentide[53] ice sheet suggests an

Antarctic contribution of 3.6–6.5 m since $6.7 \pm 0.4$ ka. This is more compatible with the glacio-eustatic history inferred by the use of BM2016 (black squares in Fig. 8).

From the data-model comparison, two subsets of best-fitting viscosity models were identified: those with relatively low LMV (~$3 \times 10^{21}$ Pa.s) and those with values an order of magnitude higher (Supplementary Fig. 3). This result is compatible with the inversion performed by Lambeck et al.[21]. Although the optimal fits (see 'Methods' section) were obtained with a LMV of $3 \times 10^{21}$ Pa.s, we cannot rule out the existence (to 95% confidence) of greater values.

To represent the RSL reconstructions on a single plot (Fig. 7), they were corrected to remove spatial variability in the GIA signal. The variability in this signal is substantial and dependent on the adopted model parameters, particularly LMV (Supplementary Fig. 4). To illustrate the importance of this spatial variability, we show model fits to RSL changes at Christmas Island (Supplementary Fig. 2), where RSL has been fairly stable over the last 6000 years[5]. Models characterised by a larger viscosity in the lower mantle tend to produce lower mid- to late-Holocene sea levels at this location, but none of the models used in this study is able to reproduce a stable RSL trend over the entire period.

In conclusion, this study brings invaluable information regarding RSL changes due to natural processes during the mid- to late-Holocene and therefore represents an important baseline of natural variability for interpreting sea-level changes in the Anthropocene. Specifically, regional short-term (multidecadal to century) variability associated with oceanographic and atmospheric changes is constrained to lie within a decimetre in most areas. Removing the GIA contribution to the reconstructed RSL changes gives a glacio-eustatic contribution of 1.5 to 2.5 m since 5.5 kyr BP that was most likely sourced from the Antarctic ice sheet. The observations also indicate isolated events of relatively rapid RSL fall and rise (several decimetres in centuries) as well as periods of remarkable stability lasting up to ~300 years. Finally, documenting the amplitude of RSL changes in this region indicates that it is suitable for studying the coastal response to sea-level change of similar amplitude, between 0.5 and 1.5 m, to

the sea-level rise that is expected to occur before the end of the current century.

## Methods

**GPS positioning.** The topographic survey of the studied outcrops is based on several thousands of measurements per study site and the measurement of the sample elevation with reference to sea level using a real-time kinematic GPS Trimble R8. The GPS is composed of a Global Navigation Satellite System (GNSS) receiver and a mobile antenna. The receiver is positioned for several hours in the field to acquire satellite data. The sample position is measured using the mobile GPS device, which is permanently radio-linked to the GNSS receiver. The maximum vertical ($Z$) and horizontal ($X$ and $Y$) elevation errors are of $\pm 2.0$ cm and a few millimetres, respectively.

During the measurement, the surveys were related to the French Polynesian Geodetic Network (Réseau Géodésique de Polynésie Française; RGPF), to operating tide gauges or tide gauge datasets, to probes that were deployed during the field work, to the instantaneous sea level or to modern adjacent microatolls growing in a similar environment than their fossil counterparts. In the absence of geodetic datum or tide gauges, probes were deployed for four to five days in order to measure the sea-level position and to compare the data to the elevation of modern microatolls. The relative sea-level curve, which is presented in this paper, is based on data acquired on islands for which longer tidal records and geodetic data are available.

After acquisition, the raw data were processed with the aims to estimate the elevation of individual dated fossil microatolls based on local tide gauge parameters, and to compare the elevation of all dated fossil microatolls according to the same vertical reference.

The link between tide gauge data and the position of the living and fossil microatolls can be established using RGPF. However, a topographic reference at the scale of French Polynesia (4167 km²), which is mandatory to achieve the second objective, does not exist, as tide gauge observations are incomplete and the NGPF (Nivellement Général de Polynésie Française) vertical datum that is associated to the RGPF is not homogeneous at this regional scale.

The official geodetic system in French Polynesia is the RGPF, which is associated with the NGPF vertical datum. The French Polynesian Geodetic Network is a semi-dynamic system with different levels established by the Naval Hydrographic and Oceanographic Service (Service Hydrographique et Océanographique de la Marine; SHOM) in cooperation with the National Geographic Institute (Institut Géographique National; IGN). The first is the reference network of French Polynesia (RRPF), which is composed of 13 sites equipped with DORIS (Doppler Orbitography by Radiopositioning Integrated by Satellite) stations that determine the position in the International Terrestrial Reference System (ITRS), specifically the ITRF92 (93.0). These 13 stations belong to a reference network from which local densifications have been conducted. A denser network consists of the deployment and the positioning of the RGPF landmarks for all Polynesian islands by the Department of Urbanism from French Polynesia (Service de l'Urbanisme de Polynésie Française). The descriptive sheet of each landmark includes its geographical coordinates that are based on the IAG-GRS80 ellipsoid and the Cartesian RGPF coordinates projected in UTM, and its elevation compared to MSL, which is determined in different ways on the various islands. In most cases, the tide gauge records are short, i.e., 1–2 weeks.

The GPS does not measure an absolute elevation but a height with respect to the reference ellipsoid, which results from mathematical modelling. Different ellipsoid models exist and the main models are the IAG-GRS80, the WGS84 and the international 1924. For all GPS surveys, the ellipsoidal height was measured compared to the mathematical model WGS84 or the Polynesian geodetic system related to the ellipsoid IAG-GRS80. The ellipsoidal heights could be compared as the SHOM recommends a difference of 10 cm between the WGS84 and the ITRF92 (IAG-GRS80 ellipsoid) models, in agreement with the set up of the geodetic network in French Polynesia.

Finally, the difference between the elevation, which is the height above the geoid, and the ellipsoid could be corrected using the global geoid models EGM96 or the more recent EGM2008 depending on the studied islands. The CIRCÉ software, produced by the IGN, was used for data processing. These global models do not take local deformations into account but the accuracy of the models grid used in this software is equal to, or lower than 4 km.

If the absence of a homogeneous altimetric network does not allow a direct comparison between the different islands, the acquisition of GPS data through a local network allows both the comparison between different study sites from the same island and the comparison of the measurements to tide gauge data.

We have determined a biological elevation reference for each study site at which a statistical representative panel of living coral microatolls was available. In total, the elevation of 126 living microatolls was measured at 12 sites from six islands with about 1500 GPS data. The study sites were selected based on the vicinity of fossil and modern microatoll fields, i.e., within a 10–600 m range. This sampling strategy provides a reproducible method and ensures the direct comparison of fossil and modern microatolls, which have lived under similar hydrodynamic and environmental conditions. For each study site, the mean biological level (MBL) was determined by calculating the average elevation of the living microatolls. It is therefore possible to compare the position of fossil microatolls from all islands, not

through their measured elevations but through the height differences between their elevation and the relevant MBL. In total, 10 MBL were determined with a margin of error of less than 3 cm.

For Bora Bora and Tikehau, the tide gauge data are robust enough to allow the comparison of the MBL based on the measurements of living microatolls to key tide gauge parameters produced by the SHOM. In Bora Bora, the MBL ranges from 27 to 5 cm bmsl depending on the reef environment (Fig. 3), while in Tikehau, the MBL is at 18 cm bmsl. No tide gauge record is available in Maupiti but the MBL was compared to the MSL recorded by the probes that we deployed. The MBL in Maupiti reef flat is 29 cm bmsl that was recorded during three days (from 5 to 8 May, 2014). This short duration explains the large deviation between the two values. The calculated MBL values are systematically below the MSL measured by the SHOM and the MSL recorded for a few days with the deployed probes. This justifies partially the use of the MBL as a robust reference for the study of fossil microatolls under the assumption of unchanged environmental and hydrological conditions through time.

In addition to the measurements of the microatoll elevations that are used in the present study, the numerous additional measured points were used to generate high-resolution maps and transects for each study site using ArcGIS 10.1 in order to reconstruct the accurate morphology of the studied outcrops. In addition, structure-from-motion photogrammetry was used to better understand the morphology and evolution of the reef system through time. High-resolution 3D models of the study sites (Figs. 2 and 4) were produced with the Agisoft Photoscan Professional software (version 1.2).

**Sample collection and dating.** The selection of microatolls for dating has been based on the lack of erosion features, the absence of local moating effects and their mineralogical preservation, demonstrating that our database is robust. The chemical preparation, mass-spectrometer measurements and age dating were performed in the years 2014–2016 mostly directly after field collection. The data are presented in Supplementary Table 2 following recommendations from Dutton et al.[54] The best-preserved samples, as indicated by X-ray powder diffraction (XRD) measurements, comprise 97.5% aragonite on average ($n = 281$). Additionally, no secondary aragonite or calcite crystals were revealed by thin section and scanning electron microscope (SEM) observations. Uranium series measurements of coral ages were performed at the mass-spectrometer facilities of the GEOMAR, Helmholtz Centre for Ocean Research Kiel, Germany. Separation of uranium and thorium from the sample matrix was done using Eichrom-UTEVA resin followed previously published methods[55]. Determination of uranium ($^{233}$U, $^{234}$U, $^{236}$U, $^{238}$U) and thorium ($^{229}$Th, $^{230}$Th, $^{232}$Th) isotope ratios was done using the multi-ion-counting inductively coupled plasma mass spectroscopy (MC-ICP-MS) approach using the method of Fietzke et al.[55] The ages were calculated using the half-lives published by Cheng et al.[56] For isotope dilution measurements, a combined $^{233}$U/$^{236}$U/$^{229}$Th spike was used with stock solutions calibrated for concentration using NIST-SRM 3164 (U) and NIST-SRM 3159 (Th) as a combi-spike, calibrated against CRM-145 uranium standard solution (formerly known as NBL-112A) for uranium isotope composition and against a secular equilibrium standard (HU-1, uranium ore solution) for the precise determination of $^{230}$Th/$^{234}$U activity ratios. Whole-procedure blank values of this sample set were measured between 0.5 and 1 pg for thorium and between 10 and 20 pg for uranium. Both values are in the range typical of this method and the laboratory. The data show that $^{238}$U concentrations range from ~2.3 ppm (RAN-44) to 3.8 ppm (TAA-11), with a mean $^{238}$U concentration of ~2.8 ppm. The concentrations of $^{232}$Th vary from 1225 ppb (TIK-74b) to 0.008 ppb (MAU-121) with an average value of about 16 ppb. Both the measured $^{232}$Th and $^{238}$U values are in typical range for young corals from oceanic islands[57–60]. The $\delta^{234}$U0 values show lowest values for sample MAU-135 of $137 \pm 0.006$‰ and highest value of $153 \pm 0.005$‰ for sample TIK-85 (Supplementary Fig. 5). Taking the $\delta^{234}$U seawater isotope value into account, it is obvious that most of the $\delta^{234}$U0 values fall within their statistical uncertainties in the range of the presently most precise $\delta^{234}$U seawater value of $146.8 \pm 0.1$‰[61]. Thirty-one samples out of 78 are slightly but significantly above or below the currently expected value in average by about 5%. This deviation from the expected value probably suggests a marginal open system behaviour of these samples. A difference of 1‰ in the $\delta^{234}$U value is expected to change the age of a 4.5 kyr BP old coral in the order of about five years. Hence, the average absolute difference of those samples slightly off the accepted value results in an age uncertainty of about $\pm 25$ years. Latter value is within the age uncertainty of our samples being in the order of about 20–30 years and can therefore be considered to be negligible.

**Relating sea-surface temperature change to sea-level change.** Output from three atmosphere-ocean general circulation models used in the CMIP5 / PIMP3 projects was used to locally evaluate the relationship between sea-surface temperatures and steric sea level at Bora Bora. Specifically, we considered output from the 'midHolocene' experiment using the MIROC-ESM and MRI-CGCM3 models, and the 'piControl' experiment using the IPSL-CM5A-MR model. Sea-surface temperature (SST; from gridded variable 'tos') and sea-surface height (SSH; from gridded variable 'zos' and global variable 'zostoga') were extracted. A 100-year running mean was applied to each time series (SST and SSH), and three pairs of

local minima or maxima were then considered in each time series in order to determine correlation coefficients between SSH and SST.

**GIA model description**. A GIA model was used to quantify the contribution of this process to the temporal and spatial variability of sea-level changes in French Polynesia. The following is a brief description of the RSL model that was used in this study, and the calibration process that was performed to identify a set of model parameters that produce optimal fits to the dataset.

The main component of the RSL model used in this study is a sea-level calculation code that computes RSL changes induced by deformation of the Earth and redistribution of ice-ocean masses. Based on the original sea-level equation by Farrell and Clark[62], the algorithm of the sea-level code is the more recent version described by Kendall et al.[63]. It includes the influence of GIA-induced changes on Earth rotation[64,65] and an improved treatment of time-dependent shoreline migration and marine-based ice growth or ablation[66]. The two main inputs to this code are an ice model describing the evolution of ice sheets over time, and an Earth model to predict the Earth's isostatic response to a loading history.

Two ice models were considered: ICE-5G[49] and that of Bradley et al.[50] (henceforth BM2016). ICE-5G was developed using a variety of observational constraints from both far-field and near-field locations[49]. BM2016 is a revised version of the ICE-3G model[67] calibrated to far-field sea-level records over the last glacial and Holocene periods[50,68]. Both ice models are provided with a time resolution of typically 0.5 or 1 kyr. This relatively coarse temporal resolution is adapted to capture millennial scale and longer changes over the most recent glacial cycle (~120 kyr ago to present), but does not capture lower-amplitude, higher-frequency changes in ice volume.

The model Earth is a spherically symmetric, compressible, Maxwell body whose response to surface loading is governed by a set of viscoelastic Love numbers[69]. Its density and elastic properties are defined by the preliminary reference Earth model[70] and its viscous properties are varied in order to improve the data-model fit. Viscosity–depth paramerisation is specified by three free parameters: lithospheric thickness (LT), upper mantle viscosity (UMV) and lower mantle viscosity (LMV).

**GIA model calibration**. The purpose of the model calibration is to determine a set of parameters that best matches the observational data. Regarding Earth model viscosity structure, the following parameter space has been explored: LT from 71 to 120 km; UMV from 0.05 to $5 \times 10^{21}$ Pa.s; LMV from 1 to $50 \times 10^{21}$ Pa.s. Each possible parameter set was combined with each of the two considered ice models (ICE-5G and BM2016) for a total of 486 model runs. In addition, ICE-5G was tested with its preferred Earth model characterised by a LT of 90 km and the depth-dependent viscosity profile VM2[49].

For each run, model results were compared to the observational data from French Polynesia. Misfit value $\chi^2$ have been calculated using a methodology[65] that incorporates both age and height uncertainty in the sea-level index points. In this method, the 'distance' $DM_{m,i}$ of the $i$th point of the dataset to the model curve $m$ is computed numerically by varying time $t$ until the following function $D_{m,i}(t)$ is minimised, as expressed by the Eq. (1) below:

$$D_{m,i}(t) = \left(\frac{Hmdl_m(t) - Hobs_i}{\sigma H_i}\right)^2 + \left(\frac{t - Tobs_i}{\sigma T_i}\right)^2 \quad (1)$$

where $i$ is the index of the observed point, $Hobs_i$ and $Tobs_i$ are the observed elevation and age of the $i$th point, $Hmdl_{m,i}$ is the predicted elevation at time $t$, $\sigma H_i$ and $\sigma T_i$ are the elevation and age errors associated with the $i$th point. Therefore, the data-model misfit $\chi^2$ for a given model $m$ is given by the Eq. (2) below:

$$\chi^2 = \frac{1}{N}\sum_{i=1}^{N} DM_{m,i} \quad (2)$$

where $i$ is the index of the observed point, $DM_{m,i}$ is the minimum of the function $D_{m,i}(t)$ as previously defined, and $N$ is the total number of observed points. It is important to note that model output was produced for the locality of each sea-level index point, and so a correction for spatial variability in this signal (as applied to generate Fig. 5) was not necessary.

Results indicate a preference for average viscosity values in the upper mantle and either 'low' or 'high' viscosity values in the lower mantle (Supplementary Fig. 3). The existence of two sets of solutions depending on LMV is consistent with the results of the inversion recently performed by Lambeck et al.[21]. The lowest $\chi^2$ values are achieved with the following models: BM2016; LT = 120 km; UMV = $10^{21}$ Pa.s; LMV = $3 \times 10^{21}$ Pa.s (Fig. 7a, black curve), and ICE-5G; LT = 71 km; UMV = $0.5 \times 10^{21}$ Pa.s; LMV = $3 \times 10^{21}$ Pa.s (Fig. 7b, black curve). These two models provide equivalent fits at the 95% confidence level, with $\chi^2$ values of 3.87 and 4.24, respectively.

**Spatial variability and GIA data correction**. Predictions of relative sea level in the South Tropical Pacific at 3 kyr BP are shown in Supplementary Fig. 4. According to the best-fitting model derived from Bradley et al.[50], the region is dominated by a long-wavelength NNE–SSW gradient in RSL. Four thousand years ago, sea level in

Christmas Island (white circle) was lower than in French Polynesia (coloured squares) that sits on a ridge of positive RSL values. Decomposing the modelled signal into that associated with ice loading, ocean loading and rotational changes shows that this regional gradient is associated with the ice-loading signal (which is dominated by the ancient Laurentide ice sheet in the tropical Pacific Ocean).

As a consequence of the spatial variability illustrated in Supplementary Fig. 4, the elevation of samples of similar age but from remote localities can differ significantly, and this difference tends to increase with the age of the samples. Thus, in order to plot RSL data from various sites on a single curve, it is necessary to remove this GIA signal by choosing a reference location and applying a model correction relative to this location. Bora Bora was chosen as a suitable reference location since the majority of observations are from this island. The largest model corrections were applied to the data from Tikehau, which exceed 10 cm prior to 5 kyr BP.

The amplitude of the abrupt sea-level rise around 5 kyr BP inferred from the GIA-corrected data, is sensitive to the adopted Earth model parameters, especially LMV. Considering a subset of models providing an equivalent fit to the dataset at the 95% confidence level, the value of the GIA correction at 5 kyr BP ranges from +5 cm to −18 cm (Supplementary Fig. 6). Thus, we infer a sea-level rise of 20 cm to 43 cm between 5.12 and 4.95 kyr BP, and rates of 1.2–2.5 mm per year.

**Data availability**. The data that support the findings of this study are available from the corresponding author on reasonable request.

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

## Acknowledgements

We thank the Polynesian authorities and services for their constant support and assistance, and all the Polynesian companies and friends for their help during field work studies. G.A.M. acknowledges support from the Natural Sciences and Engineering Research Council of Canada. E.S. acknowledges support from the Swiss National Science Foundation (Project 140618). This paper is a contribution to the PAGES/INQUA funded PALSEA2 working group.

## Author contributions

G.C. designed the study; N.H., G.C., A.E., C.V., E.S., G.A.M. and A.B. conducted field work; N.H., G.C., G.A.M. and A.B. wrote the first version of the manuscript; A.E., E.S. and C.V. contributed to writing the manuscript; N.H. compiled the database, analysed data and performed X-ray diffraction analyses; A.E. and J.Fietzke performed U-Th dating of coral samples; G.A.M. and A.B. performed geophysical modelling simulations; E.S. produced data on coral growth; C.V. performed the GPS measurements; V.P., P.D. and J.

Fleury contributed to the analysis of the GPS data; V.P. created the high-resolution maps and did the photogrammetry

## Additional information

**Competing interests:** The authors declare no competing financial interests.

