## [Peer Review File · Nature Communications]

Reviewers' comments:

Reviewer #1 (Remarks to the Author):

Comments for Hallman et al. Ice volume and climate changes from a 6,000 year sea-level record

Please include line numbers in any subsequent revision (and future manuscript) – makes the review process easier.

This is an excellent paper with an outstanding data set and state-of-the art modeling of glacial isostatic adjustment processes to derive an important record of global sea level over the last 6,000 years. As stated by the authors, such a data set is critical for assessing the background state of sea level with regard to anthropogenic changes. I strongly recommend publication in Nature Communications following modest revision.

Abstract – 3rd sentence 'Our curve...' – this is not important information and can be deleted from the abstract.

Abstract – last sentence – delete "mass" after 'ice sheet'

Abstract – last sentence – delete "remarkable" – no need for such superlatives. Let the reader decide if this is the case.

Background – first sentence – reword from "ice sheets respond to ongoing greenhouse warming" to "ice sheets respond to warming from greenhouse gas forcing."

Background – ref. 1 is wrong reference – assume it's meant to be Church et al., 2013, IPCC?

Background – 1st paragraph – "long-term" – clarify what you mean by this.

Background – 2nd paragraph – "Mid- to Late Holocene records" – provide some representative references.

Background – 3rd paragraph – "away from the major glacial centres." Rephrase "former and present ice sheets"

Background – 3rd paragraph – "the reconstructed sea-level changes" – delete "the"

Background – 3rd paragraph – "were leading to a fall in global mean sea level" – change "were leading" to "led." Also, this does not cause a change in global mean sea level.

Next section – 2nd paragraph – delete "continental"

Next section – 3rd paragraph – "studied islands does not exceed 25 cm and is not more" – change "does" to 'do' and "is" to "are" to be consistent with noun "corrections"

Need to define more clearly "hoa" and "motus" – or delete them, as not sure they are needed.

Section on Mid- to Late Holocene sea-level curve – throughout text, when using expression "Mid- to Late-Holocene" – hyphenate "Late-Holocene"

Replace

Section on Mid- to Late Holocene sea-level curve – first sentence - Our sea-level reconstruction relies on the coupling between absolute U/Th dating. Replace "relies" with "is based on" and delete "the coupling between absolute"

Section on Mid- to Late Holocene sea-level curve – reporting ages to three significant digits? I don't think this is convention? Later only report to one.

Section on Mid- to Late Holocene sea-level curve – “thus encompassing the Mid- to Late Holocene time windows” since it ends at ~1.3 ka, does not encompass the Late Holocene – only part of it.

Section on Mid- to Late Holocene sea-level curve – “referred” should be “referenced”

Section on Mid- to Late Holocene sea-level curve – (GIA corrections are applied in the following section) – not clear that Fig. 3 includes GIA corrections for different islands.

Section on Mid- to Late Holocene sea-level curve - the tidal cycle have remained – change “have” to “has”

Section on Mid- to Late Holocene sea-level curve – “and ice mass increase in the North Atlantic” - I don't think you mean “in” the North Atlantic.

Section on Mid- to Late Holocene sea-level curve - thus indicating that the MLWS was located at +70 to 80 cm between 5 and 4.6 kyr BP, thus indicating – use “thus indicating” in same sentence twice. Revise – break sentence into two?

Section on Mid- to Late Holocene sea-level curve - This abrupt rise in sea level may be related to an increase in solar irradiance at this time and a reversal of the processes hypothesised above. – Ad hoc and speculative – delete.

Section on Mid- to Late Holocene sea-level curve - The reported rates of sea-level rise in the 5.5-3.9 kyr BP time window range from ~0.1 to ~3 mm yr⁻¹ and are significantly lower than those reported from the previous interglacial period⁴¹ with rates reaching 16 mm yr⁻¹. You should be very cautious in accepting these LIG rates – they are not replicated in the two cores reported. I strongly recommend deleting this.

Section on Mid- to Late Holocene sea-level curve - Our results also contradict a long-lived Holocene sea-level highstand that was inferred from earlier studies (i.e. 5 to 1.5 kyr¹³ and 5.4 to 2 kyr BP¹⁶). – You should attempt to explain why these differences might exist.

Section on Mid- to Late Holocene sea-level curve - A sea-level fall, at an average rate of ~0.3 mm yr⁻¹, is recorded between 3.6 and 1.2 kyr BP when sea level was about a decimeter from its present position – here and in next sentence, should refer to (and compare to) Kopp's 2016 (PNAS) reconstruction for last 2 kyr.

Section on Mid- to Late Holocene sea-level curve – paragraph starting with “Our French Polynesian record displays striking differences” – should move to be first paragraph of next section.

Section on Geophysical modelling - We include, for comparison, the glacio-eustatic model inferred from a recent global analysis²⁰ – should compare here or in later discussion to reconstructions derived by Cuzzone et al. (2016) and Ullman et al. (2016), on which co-author Milne is a co-author.

Section on Geophysical modelling - While thermal expansion of the oceans over the last deglaciation was negligible in comparison to the influx of freshwater from melting ice sheets, this was not necessarily the case during – it would be relatively straightforward to put some reasonable constraints on what thermal expansion can possibly contribute.

Reviewer #2 (Remarks to the Author):

Review of Hallmann et al Ice volume and climate changes from a 6,000 year sea-level record. Hallmann et al provide a substantial new dataset of sea level proxy data from French Polynesia. These data complement existing data globally and are a substantial improvement on existing Holocene sea level records for this region, both in terms of age and elevation accuracy. These data are used to: suggest 1.5-2.5m of global ice volume sea level rise after 5.5ka; and secondly to provide a record of sea level stability/variability over the Mid-Late Holocene.

I am not convinced that these data provide any new insights into the first of these objectives because the uncertainties in the GIA approach choosing between competing ice models dose not add much to what we already know about the end of the last deglaciation. The variability vs stability aspect of the interpretation is much more robust and while there are still concerns regarding the presentation of the data and the interpretation I would recommend that this would be suitable for publication and would be of interest to a wide readership.

I recommend that the authors amend the manuscript and reduce the focus on the GIA modelled global sea level change and focus more on the variability of sea level within the record they have and widen the discussion here. Additionally the authors must provide some archive of both the elevation filed data and the U-Th data.

Local sea level variability

Augments about tectonic stability are sound given the short time frame of this study

The heights of modern living micro atolls are used to constrain the accuracy of the sea level estimates as well as the offset between mean sea levels and the height of micro atoll growth. Ranges and means are provided in the text but there is no data table of these measurements provided. Such a table should include the heights, and localities as well as the tidal ranges at each site. A similar data table for the elevations and localities of the samples used in this study is not provided. (see call for data tables below).

There seems to be a contradiction in the data between the inference of stability in sea level from the large microatolls and the other samples. For example during the period from 5.5 to 5.1 kyr BP the large microatoll suggests a period of sea level stability whereas the other data show a fall in sea level of 20cm at exactly the same time. This suggest that there is some additional uncertainty in the sea level interpretation. The text states "Moreover, the occurrence of very large microatolls in this field indicates a stable sea-level position for several centuries after the abrupt drop" but from the figure it looks like this large microatoll data spans the drop and not postdates it. This could be an issue with my interpretation of the figure.

Additionally there is a contradiction between the statement that the large microatolls indicate a period of sea level stability while the microtopography within these atolls is then used to suggest that there is higher frequency sea level oscillations**. While these are statements of different timescales, of 100s of years of stability versus sub-decadal, the inferred magnitudes of change are somewhat similar. It would be worth more explanation here of how these records are recording different modes of sea level variability.

**oscillations is not the correct term as it infers a periodic variation around a mean rather than more simply variability.

What is the magnitude of a steric sea level change? This is proposed as an explanation of the fall in sea level at 6-5 kyrBP. How much did the ocean cool and how much would this affect sea level? The authors mention this possibility* but it would be good to attempt to put some quantitative constraints on this. Just roughly considering the IPCC numbers for RCP 2.6 at 2100 a there will be a 1C warming and the thermal expansion component of sea level rise would be about 15cm. This is almost enough to explain all of the short term variability in the record.

*Meehl et al 2009 is not an appropriate reference for the statement that there was a cooling in the Pacific in the mid Holocene. This reference can explain how a change in forcing might be amplified but the timescales are different and it does not provide evidence of an actual change in temperature at this time. Stott et al Nature 431, 56-59 (2004) does however show some SST variability at this time of about 1C which might be significant.

The rise in sea level inferred at 5-4.6 kyr BP is between different sites and so the authors have had to rely on the accuracy of the GIA correction to accurately constrain this change. This approach is sound provided the error in this correction adequately documented.

The findings that the amplitude of the Holocene sea level highstand is much less than in previous studies (Pirazzoli & Montaggioni 1988 and Rashid et al 2014) needs further explanation. The shorter duration can be explained by improvements in dating accuracy but the lower elevation of the samples cannot. The authors need to explain the higher elevation samples of the earlier work are not accurate sea level markers.

GIA modelling and global glacio-eustatic sea level

The GIA modelling appears to contain an element of circularity. Where the GIA signals are removed to estimate a global ice volume change this requires the use of the ice models which in turn incorporate a prior of the ice volume history. This is why the data and models agree in figure 6 but not between each ice model used. The value of ice melting given in the text of ~1.5 m to ~2.5 m since ~5,500 years BP is a function not of the data used here but of the ice models used to do the GIA corrections. In this case the difference between the ice models is about half the magnitude of the total change. The range of 1.5 -2.5 m is just the range from using two different ice models. If a third ice model were used this range could be even larger. It seems that the largest uncertainty in knowing what the glacio-eustatic contribution was from 5.5 kyrBP to present is from the ice model used and not from the proxy data. It is not clear that this manuscript adds much more in the way of certainty in choosing between ice models or in containing global ice volume on this timescale.

Figures I think need improvement

Figure 2 the photogrammetry is informative as to the sampling site but could be improved by colour shading the topography with the elevation instead/as well as the draped images this would show better the range of heights that the micro atolls constrain sea level. Also is there such photogrammetry for the large microatolls used to infer long still stands? These would be good to include either here or in the SI.

Figure 3 it is not clear what is meant by the "plateaus". I think they might be referring to the data points that are extended horizontally such as the wide data points of Tikehau (purple). Some additional legend or different symbols for the large microatolls here would be useful. Throughout the text there are references to rises and falls of sea level and still stands indicated but large microatolls. These are really hard to pick out from this figure. I would suggest that there is a panel that includes an interpretive sea level curve or at least something to highlight where sea level is locally rising, falling, or is stable. The Holocene reef flat data should be made more obviously different than the microatoll data (open symbols?) and if possible put onto fig 3a as well.

Data.

There is a lack of data archiving for this manuscript. I would like to see the coral elevation data and site metadata archived in some assessable way. This should include the localities of the sites, their elevation and how that elevation was established, the local measured tidal ranges, and the elevation of the inferred sea level proxy point from each site, including uncertainties. The U-Th data also needs to be presented in such a way that ages can be recalculated if required (<https://doi.org/10.1016/j.quageo.2017.03.001>).

Response to reviewers' comments

Manuscript NCOMMS-17-12160-T

"Ice volume and climate changes from a 6,000 year sea-level record"

Reviewer #1:

Line numbers were included.

Abstract – 3rd sentence ‘Our curve...’ – this is not important information and can be deleted from the abstract.

We kept this sentence in the abstract as we think that it contains crucial information. We think that stating the number of sea-level index points is important and that this study is unique in analyzing microatolls from several islands in a single region.

Abstract – last sentence – delete “mass” after ‘ice sheet’

This sentence was deleted.

Abstract – last sentence – delete “remarkable” – no need for such superlatives. Let the reader decide if this is the case.

Remarkable was not meant in the sense of quality of the data but to emphasize the significance of these periods of sea-level stability. We replaced 'remarkable' by 'significant'.

Background – first sentence – reword from “ice sheets respond to ongoing greenhouse warming” to “ice sheets respond to warming from greenhouse gas forcing.”

Wording was changed accordingly.

Background – ref. 1 is wrong reference – assume it’s meant to be Church et al., 2013, IPCC?

The reference was corrected.

Background – 1st paragraph – “long-term” – clarify what you mean by this.

The word 'long-term' was clarified. 'Long-term' was changed to 'centennial to millennial'.

Background – 2nd paragraph – “Mid- to Late Holocene records” – provide some representative references.

Two references were added: Gehrels (1999) and Woodroffe et al. (2012).

Background – 3rd paragraph – “away from the major glacial centres.” Rephrase “former and present ice sheets”

The wording was changed.

Background – 3rd paragraph – “the reconstructed sea-level changes” – delete “the”

'The' was deleted.

Background – 3rd paragraph – “were leading to a fall in global mean sea level” – change “were leading” to “led.’ Also, this does not cause a change in global mean sea level.

'Leading' was changed to 'led'.

'Global mean' was deleted.

Next section – 2nd paragraph – delete “continental”

'Continental' was deleted.

Next section – 3rd paragraph – “studied islands does not exceed 25 cm and is not more” – change “does” to ‘do” and “is” to “are” to be consistent with noun “corrections”

Words were corrected accordingly.

Need to define more clearly “hoa” and “motus” – or delete them, as not sure they are needed.

The word 'hoa' is already defined. 'Motus' were defined in the text. We think that these two expressions are needed as they are classical terms for reef morphology.

Section on Mid- to Late Holocene sea-level curve – throughout text, when using expression “Mid- to Late-Holocene” – hyphenate “Late-Holocene”

The spelling was changed based on other *Nature Communication* papers:

Changed 'Late Holocene' to 'late-Holocene';

changed 'Mid-Holocene' to 'mid-Holocene'.

Section on Mid- to Late Holocene sea-level curve – first sentence - Our sea-level reconstruction relies on the coupling between absolute U/Th dating. Replace “relies” with “is based on” and delete “the coupling between absolute”

The changes were done accordingly.

Section on Mid- to Late Holocene sea-level curve – reporting ages to three significant digits? I don't think this is convention? Later only report to one.

U-series ages were recalculated and only 2 instead of 3 digits are reported in the revised version.

Section on Mid- to Late Holocene sea-level curve – “thus encompassing the Mid- to Late Holocene time windows” since it ends at ~1.3 ka, does not encompass the Late Holocene – only part of it.

Wording was changed to 'most of the mid- to late-Holocene'.

Section on Mid- to Late Holocene sea-level curve – “referred” should be “referenced”

The word was changed.

Section on Mid- to Late Holocene sea-level curve – (GIA corrections are applied in the following section) – not clear that Fig. 3 includes GIA corrections for different islands.

'GIA corrections were not applied to the data.' was added to Figure 3 caption.

Section on Mid- to Late Holocene sea-level curve - the tidal cycle have remained – change “have” to “has”

Wording was changed.

Section on Mid- to Late Holocene sea-level curve – “and ice mass increase in the North Atlantic” - I don't think you mean “in” the North Atlantic.

Was changed to 'North Atlantic region'

Section on Mid- to Late Holocene sea-level curve - thus indicating that the MLWS was located at +70 to 80 cm between 5 and 4.6 kyr BP, thus indicating – use “thus indicating” in same sentence twice. Revise – break sentence into two?

'thus' was deleted from the sentence. The sentence was rewritten.

Section on Mid- to Late Holocene sea-level curve - This abrupt rise in sea level may be related to an increase in solar irradiance at this time and a reversal of the processes hypothesised above. – Ad hoc and speculative – delete.

The sentence was deleted.

Section on Mid- to Late Holocene sea-level curve - The reported rates of sea-level rise in the 5.5-3.9 kyr BP time window range from ~0.1 to ~3 mm yr⁻¹ and are significantly lower than those reported from the previous interglacial period⁴¹ with rates reaching 16 mm yr⁻¹. You should be very cautious in accepting these LIG rates – they are not replicated in the two cores reported. I strongly recommend deleting this.

We followed the recommendation and deleted the reference to Rohling et al. (2008). We replaced this reference by a new reference, Dutton et al. (2015), to give a range of sea-level rates for the Last Interglacial for comparison to our Holocene data.

Section on Mid- to Late Holocene sea-level curve - Our results also contradict a long-lived Holocene sea-level highstand that was inferred from earlier studies (i.e. 5 to 1.5 kyr¹³ and 5.4 to 2 kyr BP¹⁶). – You should attempt to explain why these differences might exist.

An explanation on the difference in amplitude and timing of the highstand between previous studies and our study was added. See also comment by Reviewer #2.

Section on Mid- to Late Holocene sea-level curve - A sea-level fall, at an average rate of ~0.3 mm yr⁻¹, is recorded between 3.6 and 1.2 kyr BP when sea level was about a decimeter from its present position – here and in next sentence, should refer to (and compare to) Kopp's 2016 (PNAS) reconstruction for last 2 kyr.

Kopp et al. (2016) reconstructed global sea-level change. Thus, a comparison to our regional curve appears inappropriate.

Section on Mid- to Late Holocene sea-level curve – paragraph starting with “Our French Polynesian record displays striking differences” – should move to be first paragraph of next section.

As suggested, the paragraph was moved to the beginning of the next section.

Section on Geophysical modelling - We include, for comparison, the glacio-eustatic model inferred from a recent global analysis²⁰ – should compare here or in later discussion to reconstructions derived by Cuzzone et al. (2016) and Ullman et al. (2016), on which co-author Milne is a co-author.

The results of Ullman et al. (2016) have been worked into the discussion as they provide

useful constraints on the Antarctic contribution from ~6.7 ka. This constraint is more compatible with our glacio-eustatic history inferred from the BM2016 ice model.

Section on Geophysical modelling - While thermal expansion of the oceans over the last deglaciation was negligible in comparison to the influx of freshwater from melting ice sheets, this was not necessarily the case during – it would be relatively straightforward to put some reasonable constraints on what thermal expansion can possibly contribute.

We have added more details in the section ‘Mid-to-late Holocene sea-level curve’ on the correlation of sea-surface height change with sea-surface temperature change based on examining output from AOGCMs. Based on this correlation, we have added information on the plausibility of a steric/dynamic signal in causing some of the isolated, rapid events identified as well as a more secular contribution to RSL during the mid-to-late Holocene (the latter discussed in the section ‘Geophysical modelling’).

Reviewer #2:

I am not convinced that these data provide any new insights into the first of these objectives because the uncertainties in the GIA approach choosing between competing ice models dose not add much to what we already know about the end of the last deglaciation. The variability vs stability aspect of the interpretation is much more robust and while there are still concerns regarding the presentation of the data and the interpretation I would recommend that this would be suitable for publication and would be of interest to a wide readership.

I recommend that the authors amend the manuscript and **reduce the focus on the GIA modelled global sea level change and focus more on the variability of sea level within the record** they have and widen the discussion here. Additionally the authors must **provide some archive of both the elevation filed data and the U-Th data.**

The focus on the GIA modelled global sea-level change has shifted more to the variability of sea level within the record. Discussion and data on modern sea-level variability were added. Apparent variability within our Holocene record was addressed (see paragraph 'Local sea-level variability' below).

The lack of data archiving was addressed (see paragraph 'Data' below).

Local sea level variability

Augments about tectonic stability are sound given the short time frame of this study

The heights of **modern living micro atolls** are used to constrain the accuracy of the sea level

estimates as well as the offset between mean sea levels and the height of micro atoll growth. Ranges and means are provided in the text but there is no data table of these measurements provided. Such a table should include the heights, and localities as well as the tidal ranges at each site. A similar data table for the elevations and localities of the samples used in this study is not provided. (see call for data tables below).

Supplementary Table 1 was added and includes sample heights and localities of fossil microatolls; Supplementary Table 2 summarizes the U/Th data. A new figure was added (Fig. 2) to illustrate the relationships between modern microatoll development and tidal parameters within different reef environments.

There seems to be a contradiction in the data between the inference of stability in sea level from the large microatolls and the other samples. For example during the period from 5.5 to 5.1 kyr BP the large microatoll suggests a period of sea level stability whereas the other data show a fall in sea level of 20 cm at exactly the same time. This suggest that there is some additional uncertainty in the sea level interpretation. The text states “Moreover, the occurrence of very large microatolls in this field indicates a stable sea-level position for several centuries after the abrupt drop” but from the figure it looks like this large microatoll data spans the drop and not postdates it. This could be an issue with my interpretation of the figure.

The sea-level curve (Fig. 5) was revised as the length of the bar of sample TIK-74 was initially exaggerated.

Additionally there is a contradiction between the statement that the large microatolls indicate a period of sea level stability while the microtopography within these atolls is then used to suggest that there is higher frequency sea level oscillations**. While these are statements of different timescales, of 100s of years of stability versus sub-decadal, the inferred magnitudes of change are somewhat similar. It would be worth more explanation here of how these records are recording different modes of sea level variability.

**oscillations is not the correct term as it infers a periodic variation around a mean rather than more simply variability.

Explanation on the meaning of large microatolls and their microtopography was added. The microtopography reflects high-frequency, low-amplitude sea-level variations at a different timescale.

Changed the word 'oscillation' to 'variation'.

What is the magnitude of a steric sea level change? This is proposed as an explanation of the fall in sea level at 6-5 kyrBP. How much did the ocean cool and how much would this affect sea level? The authors mention this possibility* but it would be good to attempt to put some quantitative constraints on this. Just roughly considering the IPCC numbers for RCP 2.6 at 2100 a there will be a 1C warming and the thermal expansion component of sea level rise would be about 15cm. This is almost enough to explain all of the short term variability in the record.

We have considered multi-centennial outputs from 3 AOGCMs used in PIMP3 / CIMP5 experiments to determine if a quantitative relationship exists between sea-surface temperature change and sea-surface height change, using natural radiative forcing only (pre-industrial control and mid-Holocene experiments). Such a correlation has been shown to exist using modern observations (for instance, Casey and Adamec, 2002), although only decadal / subdecadal variations have been analysed due to the data available (reconstructions from tide gauge records, satellite altimetry). A paragraph has been added to the section 'Mid-to-late Holocene sea-level curve' describing the results of this exercise.

*Meehl et al 2009 is not an appropriate reference for the statement that there was a cooling in the Pacific in the mid Holocene. This reference can explain how a change in forcing might be amplified but the timescales are different and it does not provide evidence of an actual change in temperature at this time. Stott et al Nature 431, 56-59 (2004) does however show some SST variability at this time of about 1C which might be significant.

As suggested, the reference Meehl et al. (2009) was deleted and Stott et al. (2004) was added. The sentence was rephrased accordingly.

The rise in sea level inferred at 5-4.6 kyr BP is between different sites and so the authors have had to rely on the accuracy of the GIA correction to accurately constrain this change. This approach is sound provided the error in this correction adequately documented.

This is documented in the Supplementary Information section.

The findings that the amplitude of the Holocene sea level highstand is much less than in previous studies (Pirazzoli & Montaggioni 1988 and Rashid et al 2014) needs further explanation. The shorter duration can be explained by improvements in dating accuracy but the lower elevation of the samples cannot. The authors need to explain the higher elevation samples of the earlier work are not accurate sea level markers.

An explanation on the difference in amplitude and timing of the highstand between previous studies and our study was added.

GIA modelling and global glacio-eustatic sea level

The GIA modelling appears to contain an element of circularity. Where the GIA signals are removed to estimate a global ice volume change this requires the use of the ice models which in turn incorporate a prior of the ice volume history. This is why the data and models agree in figure 6 but not between each ice model used. The value of ice melting given in the text of ~1.5 m to ~2.5 m since ~5,500 years BP is a function not of the data used here but of the ice models used to do the GIA corrections. In this case the difference between the ice models is about half the magnitude of the total change. The range of 1.5 -2.5 m is just the range from using two different ice models. If a third ice model were used this range could be even larger. It seems that the largest uncertainty in knowing what the glacio-eustatic contribution was from 5.5 kyrBP to present is from the ice model used and not from the proxy data. It is not clear that this manuscript adds much more in the way of certainty in choosing between ice models or in containing global ice volume on this timescale.

Yes, there is an element of circularity and this has now been explicitly noted in the discussion that describes the results in Figure 8. We have also mentioned that the range in the inferred ice volume change is largely attributed to the differences in the ice models used to remove the GIA signal from the observations.

Figures I think need improvement

Former figures were modified according to recommendations and new figures were added (see below for details).

Figure 2 the photogrammetry is informative as to the sampling site but could be improved by colour shading the topography with the elevation instead/as well as the draped images this would show better the range of heights that the micro atolls constrain sea level. Also is there such photogrammetry for the large microatolls used to infer long still stands? These would be good to include either here or in the SI.

Figure 2 was modified accordingly and a new figure on the photogrammetry of a large microatoll was added.

Figure 3 it is not clear what is meant by the “plateaus”. I think they might be referring to the

data points that are extended horizontally such as the wide data points of Tikehau (purple). Some additional legend or different symbols for the large microatolls here would be useful. Throughout the text there are references to rises and falls of sea level and still stands indicated but large microatolls. These are really hard to pick out from this figure. I would suggest that there is a panel that includes an interpretive sea level curve or at least something to highlight where sea level is locally rising, falling, or is stable. The Holocene reef flat data should be made more obviously different than the microatoll data (open symbols?) and if possible put onto fig 3a as well.

Figure 5 (former Figure 3) caption was rewritten. The word 'plateau' was replaced by 'stillstand' in the main text and in Figure 5 caption. In addition, an explanation of the interpretation of large microatolls was added to the main text.

We decided not to use different symbols for large microatolls and not to change the legend. All microatolls are shown by the same (solid) symbol. The horizontal extension of the bars indicates the duration of the stillstand, i.e. the longer the bar, the larger the microatoll, and therefore the longer the sea-level stillstand. As suggested by Reviewer #2, a new sub-panel including our interpretation of the coral data was added to Fig. 5a.

Open symbols were used for reef flat data in Fig. 5b. Figure 5a includes exclusively data derived from microatolls, i.e. the elevation difference between modern and fossil microatolls. As this is not applicable for the reef flat, the reef flat data are only shown in Fig 5b where the measured elevation (NGPF reference) is shown.

Data.

There is a lack of data archiving for this manuscript. I would like to see the coral elevation data and site metadata archived in some assessable way. This should include the localities of the sites, their elevation and how that elevation was established, the local measured tidal ranges, and the elevation of the inferred sea level proxy point from each site, including uncertainties. The U-Th data also needs to be presented in such a way that ages can be recalculated if required (<https://doi.org/10.1016/j.quageo.2017.03.001>).

The lack of data archiving was addressed. All data, i.e. elevations of microatolls, localities, tides, U-series data (following the suggested data reporting method for publishing U-series data by Dutton et al., 2017) are part of the manuscript or the Supplementary Information. Two data tables were added to the Supplementary Information (Supplementary Tables 1 and 2) and

the tidal ranges are shown in Figure 2, which was added to the main text. Furthermore, the data of this study will be added to the global Holocene PALSEA sea-level database.

Additional changes:

Changed 'mm⁻¹' to 'mm per year' (following the formatting of other *Nature Communication* papers).

REVIEWERS' COMMENTS:

Reviewer #2 (Remarks to the Author):

This manuscript represents a substantial improvement upon the initial submission.

I am satisfied that the presentation of the chronology and elevation data are acceptable, although I would recommend that the data tables presented in the SI are also uploaded to <https://www.pangaea.de/> in addition to being archived at Nature.

The focus of the manuscript is now less towards the Ice volume equivalent sea level and is more appropriately focused on the relative sea level changes and their relevance for background natural variability upon which anthropogenic sea level rise is superimposed.

I recommend that the manuscript be published in its current form with no required alterations.

Reviewer #2:

As suggested by Reviewer #2 the two Supplementary Tables including all data were uploaded to PANGAEA (PDI-16384).